# ONE-STAGE PROMPT-BASED CONTINUAL LEARNING

## ABSTRACT

Prompt-based Continual Learning (PCL) has gained considerable attention as a promising continual learning solution as it achieves state-of-the-art performance while preventing privacy violation and memory overhead issues. Nonetheless, existing PCL approaches face significant computational burdens because of two Vision Transformer (ViT) feed-forward stages; one is for the query ViT that generates a prompt query to select prompts inside a prompt pool; the other one is a backbone ViT that mixes information between selected prompts and image tokens. To address this, we introduce a one-stage PCL framework by directly using the intermediate layer's token embedding as a prompt query. This design removes the need for an additional feed-forward stage for query ViT, resulting in $\sim 50\%$ computational cost reduction for both training and inference with marginal accuracy drop ($\leq 1\%$). We further introduce a Query-Pool Regularization (QR) loss that regulates the relationship between the prompt query and the prompt pool to improve representation power. The QR loss is only applied during training time, so there is no computational overhead at inference from the QR loss. With the QR loss, our approach maintains $\sim 50\%$ computational cost reduction during inference as well as outperforms the prior two-stage PCL methods by $\sim 1.4\%$ on public class-incremental continual learning benchmarks including CIFAR-100 and ImageNet-R.

## 1 INTRODUCTION

Training models effectively and efficiently on a continuous stream of data presents a significant practical hurdle. A straightforward approach would entail accumulating both prior and new data and then updating the model using this comprehensive dataset. However, as data volume grows, fully retraining a model on such extensive data becomes increasingly impractical (Mai et al., 2022; Hadsell et al., 2020). Additionally, storing past data can raise privacy issues, such as those highlighted by the EU General Data Protection Regulation (GDPR) (Voigt & Von dem Bussche, 2017). An alternative solution is to adapt the model based solely on currently incoming data, eschewing any access to past data. This paradigm is termed as *rehearsal-free continual learning* (Choi et al., 2021; Gao et al., 2022; Yin et al., 2020; Smith et al., 2021; Wang et al., 2022d;c; Smith et al., 2023a), and the primary goal is to diminish the effects of catastrophic forgetting on previously acquired data.

Among the rehearsal-free continual learning methods, Prompt-based Continual Learning (PCL) stands out as it has demonstrated state-of-the-art performance in image classification tasks, even surpassing rehearsal-based methods (Wang et al., 2022d;c; Smith et al., 2023a; Wang et al., 2022a). PCL utilizes a pre-trained Vision Transformer (ViT) (Dosovitskiy et al., 2021) and refines the model by training learnable tokens on the given data. PCL adopts a prompt pool-based training scheme where different prompts are selected and trained for each continual learning stage. This strategy enables a model to learn the information of the training data in a sequential manner with less memory overhead, as the prompt pool requires minimal resources.

Although PCL methods show state-of-the-art performance, huge computational costs from the two ViT feed-forward stages make the model difficult to deploy into resource-constrained devices (Harun et al., 2023; Wang et al., 2022b; Pellegrini et al., 2021). Specifically, the PCL method requires two-stage ViT feedforward steps. One is for the query function that generates a prompt query. The other one is a backbone ViT that mixes information between selected prompts and input image tokens. We refer to this approach as a *two-stage PCL* method, illustrated in Fig. 1 Left.

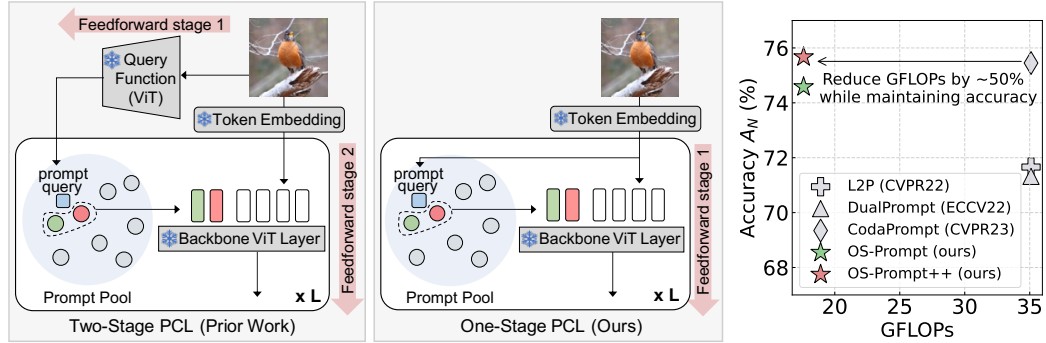

Figure 1: Difference between prior PCL work and ours. Prior PCL work (**Left**) has two feed-forward stages for (1) a query function (ViT) to select input-specific prompts and (2) a backbone ViT layer to perform prompt learning with the selected prompts and image tokens. On the other hand, our one-stage PCL framework (**Middle**) uses an intermediate layer's token embedding as a prompt query so that it requires only one backbone ViT feed-forward stage. As a result, our method reduces GFLOPs by $\sim 50\%$ compared to prior work while maintaining accuracy (**Right**).

To address this limitation, we propose a *one-stage PCL* framework with only one ViT feed-forward stage (Fig. 1 Middle), called OS-Prompt. Rather than deploying a separate ViT feed-forward phase to generate a prompt query, we directly use the intermediate layer's token embedding as a query. This is based on our observation that early layers show marginal shifts in the feature space during continual prompt learning (details are in Section 4.1). This observation enables us to use the intermediate token embedding as a prompt query which requires consistent representation across continual learning to minimize catastrophic forgetting. Surprisingly, OS-Prompt shows a marginal performance drop (less than 1%) while saving $\sim 50\%$ of computational cost (Fig. 1 Right).

As our OS-Prompt uses the intermediate layer's token embedding as a query instead of the last layer's token embedding (which is used in prior PCL methods), there is a slight performance drop due to the lack of representation power. To address this, we introduce a Query-Pool Regularization (QR) loss, which is designed to enhance the representation power of a prompt pool by utilizing the token embedding from the last layer. Importantly, the QR loss is applied only during training, ensuring no added computational burden during inference. We refer to our enhanced one-stage framework with QR loss as OS-Prompt++. Our OS-Prompt++ bridges the accuracy gap, performing better than OS-Prompt.

Overall, our contribution can be summarized as follows: (1) We raise a drawback of the current PCL methods — high computational cost, particularly due to the two distinct ViT feed-forward stages. As a solution to the computational inefficiency in existing PCL techniques, we propose OS-Prompt that reduces the computational cost by nearly 50% without any significant performance drop. (2) To counter the slight performance degradation observed in our one-stage PCL framework, we introduce a QR loss. This ensures that the prompt pool maintains similarity with token embedding from both intermediate and final layers. Notably, our QR loss avoids any additional computational overhead during inference. (3) We conduct experiments with rehearsal-free continual learning setting on CIFAR-100 (Krizhevsky et al., 2009) and ImageNet-R (Hendrycks et al., 2021) benchmarks, outperforming the previous SOTA method CodaPrompt (Smith et al., 2023a) by $\sim 1.4\%$ with $\sim 50\%$ computational cost saving.

## 2 RELATED WORK

### 2.1 CONTINUAL LEARNING

For a decade, continual learning has been explored as an important research topic, concentrating on the progressive adaptation of models over successive tasks or datasets. One of the representative methods to address continual learning is the regularization-based method (Aljundi et al., 2018; Li & Hoiem, 2017; Zenke et al., 2017). By adding a regularization loss between the current model and the previous model, these methods aim to minimize catastrophic forgetting. However, their performance is relatively low on challenging datasets compared to the other continual learning methods.

An alternative approach proposes to expand the network architecture as it progresses through different continual learning stages (Li et al., 2019; Rusu et al., 2016; Yoon et al., 2017; Serra et al., 2018). Though these typically surpass the results of regularization-based methods, they come at the expense of significant memory consumption due to the increased parameters. Rehearsal-based continual learning has introduced a component for archiving previous data (Chaudhry et al., 2018; 2019; Hayes et al., 2019; Buzzega et al., 2020; Rebuffi et al., 2017). By leveraging accumulated data during the subsequent stages, they often outperform other continual learning methods. However, saving images causes memory overhead and privacy problems. Given the inherent limitations, a growing interest is observed in rehearsal-free continual learning. These are crafted to address catastrophic forgetting without access to past data. Most of them propose methods to generate rehearsal images (Choi et al., 2021; Gao et al., 2022; Yin et al., 2020; Smith et al., 2021), but generating images is resource-heavy and time-consuming. Recently, within the rehearsal-free domain, Prompt-based Continual Learning (PCL) methods have gained considerable attention because of their performance and memory efficiency. By utilizing a pre-trained ViT model, they train only a small set of parameters called prompts to maintain the information against catastrophic forgetting. For example, L2P (Wang et al., 2022d) utilizes a prompt pool, selecting the appropriate prompts for the given image. Extending this principle, DualPrompt (Wang et al., 2022c) proposes general prompts for encompassing knowledge across multiple continual learning stages. Advancing this domain further, Smith et al. (2023a) facilitates end-to-end training of the prompt pool, achieving state-of-the-art performance. However, huge computational costs from the two ViT feed-forward stages make the model difficult to deploy into resource-constrained devices. Our work aims to resolve such computational complexity overhead in PCL.

## 2.2 PROMPT-BASED LEARNING

The efficient fine-tuning method for large pre-trained models has shown their practical benefits across various machine learning tasks (Rebuffi et al., 2018; Zhang et al., 2020; 2021; Zhou et al., 2022; He et al., 2022; Hu et al., 2021). Instead of adjusting all parameters within neural architectures, the emphasis has shifted to leveraging a minimal set of weights for optimal transfer outcomes. In alignment with this, multiple methodologies (Rusu et al., 2016; Cai et al., 2020) have integrated a streamlined bottleneck component within the transformer framework, thus constraining gradient evaluations to select parameters. Strategies like TinyTL (Cai et al., 2020) and BitFit (Zaken et al., 2021) advocate for bias modifications in the fine-tuning process. In a recent shift, prompt-based learning (Jia et al., 2022; Khattak et al., 2023) captures task-specific knowledge with much smaller additional parameters than the other fine-tuning methods (Wang et al., 2022d). Also, prompt-based learning only requires storing several prompt tokens, which are easy to plug-and-play, and hence, they are used to construct a prompt pool for recent continual learning methods.

## 3 PRELIMINARY

### 3.1 PROBLEM SETTING

In a continual learning setting, a model is trained on $T$ continual learning stages with dataset $D = \{D_1, D_2, ..., D_T\}$, where $D_t$ is the data provided in $t$-th stage. Our problem is based on an image recognition task, so each data $D_t$ consists of pairs of images and labels. Also, data from the previous tasks is not accessible for future tasks. Following the previous rehearsal-free continual learning settings (Wang et al., 2022c; Smith et al., 2023a), we focus on the class-incremental continual learning setting where task identity is unknown at inference. This is a challenging scenario compared to others such as task-incremental continual learning where the task labels are provided for both training and test phases (Hsu et al., 2018). For the experiments, we split classes into $T$ chunks with continual learning benchmarks including CIFAR-100 (Krizhevsky et al., 2009) and ImageNet-R (Hendrycks et al., 2021).

### 3.2 TWO-STAGE PROMPT-BASED CONTINUAL LEARNING

In our framework, we focus on improving efficiency through a new query selection rather than changing the way prompts are formed from the prompt pool with the given query, a common focus in earlier PCL studies (Wang et al., 2022c;d; Smith et al., 2023a). For a fair comparison with earlier work, we follow the overall PCL framework established in previous studies. Given that our

contribution complements the prompt formation technique, our method can seamlessly work with future works that propose a stronger prompt generation method.

The PCL framework selects the input-aware prompts from the layer-wise prompt pool and adds them to the backbone ViT to consider task knowledge. The $l$-th layer has the prompt pool $P_l = \{k_l^1 : p_l^1, ..., k_l^M : p_l^M\}$ which contains $M$ prompt components $p \in \mathbb{R}^{L_p \times D}$ and the corresponding key $k \in \mathbb{R}^D$. Here, $L_p$ and $D$ stand for the prompt length and the feature dimension, respectively.

The prior PCL method consists of two feed-forward stages. In the first stage, a prompt query $q \in \mathbb{R}^D$ is extracted from pre-trained query ViT $Q(\cdot)$, utilizing the $[CLS]$ token from the final layer.

$$q = Q(x)_{[CLS]}. \tag{1}$$

Here $x$ is the given RGB image input. The extracted prompt query is used to form a prompt $\phi_l \in \mathbb{R}^{L_p \times D}$ from a prompt pool $P_l$, which can be formulated as:

$$\phi_l = g(q, P_l). \tag{2}$$

The prompt formation function $g(\cdot)$ has been a major contribution in the previous literature. L2P (Wang et al., 2022d) and DualPrompt (Wang et al., 2022c) select prompts having top-N similarity (*e.g.*, cosine similarity) between query $q$ and prompt key $k_l$. The recent state-of-the-art CodaPrompt (Smith et al., 2023a) conducts a weighted summation of prompt components based on their similarity to enable end-to-end training.

The obtained prompt $\phi_l$ for layer $l$ is given to backbone ViT layer $f_l$ with input token embedding $x_l$. This is the second feed-forward stage of the prior PCL methods.

$$x_{l+1} = f_l(x_l, \phi_l). \tag{3}$$

Following DualPrompt (Wang et al., 2022c) and CodaPrompt (Smith et al., 2023a), we use prefix-tuning to add the information of a prompt $\phi_l$ inside $f_l$. The prefix-tuning splits prompt $\phi_l$ into $[\phi_k, \phi_v] \in \mathbb{R}^{\frac{L_p}{2} \times D}$, and then prepends them to the key and value inside the self-attention block of ViT. We utilize Multi-Head Self-Attention (MHSA) (Dosovitskiy et al., 2021) like the prior PCL work, which computes the outputs from multiple single-head attention blocks.

$$MHSA(x, \phi_k, \phi_v) = Concat[head_1, ..., head_H]W_o. \tag{4}$$

$$head_i = Attention(xW_q^i, [\phi_k; x]W_k^i, [\phi_v; x]W_v^i). \tag{5}$$

Here, $W_o, W_q, W_k, W_v$ are the projection matrices. This prefix-tuning method is applied to the first 5 layers of backbone ViT. The leftover layers conduct a standard MHSA without prompts.

However, the prior PCL approach requires two-stage ViT feedforward (Eq. 1 and Eq. 3), which doubles computational cost. In our work, we aim to improve the computational efficiency of the PCL framework without degrading the performance.

## 4 METHODOLOGY: ONE-STAGE PROMPT-BASED CONTINUAL LEARNING

We propose to reduce the computational cost of PCL by restructuring two-step feedforward stages. To this end, we propose a new one-stage PCL framework called *OS-Prompt*. Instead of using a separate feedforward step to compute the prompt query $q$ from Eq. 1, we take a token embedding from the intermediate layer of a backbone ViT as the query (illustrated in Fig. 1).

### 4.1 HOW STABLE ARE TOKEN EMBEDDINGS ACROSS CONTINUAL PROMPT LEARNING?

We first ensure the validity of using intermediate token embedding as a prompt query. The original two-stage design employs a frozen pre-trained ViT, ensuring consistent prompt query representation throughout continual learning. It is essential to maintain a consistent (or similar) prompt query representation because changes in prompt query would bring catastrophic forgetting. In our approach, token embeddings in the backbone ViT continually change as prompt tokens are updated during learning. For instance, after training on tasks 1 and 2, prompts have different values, resulting in varying token embeddings for an identical image. Consequently, it is crucial to assess if token embeddings sustain a consistent representation throughout the continual learning.

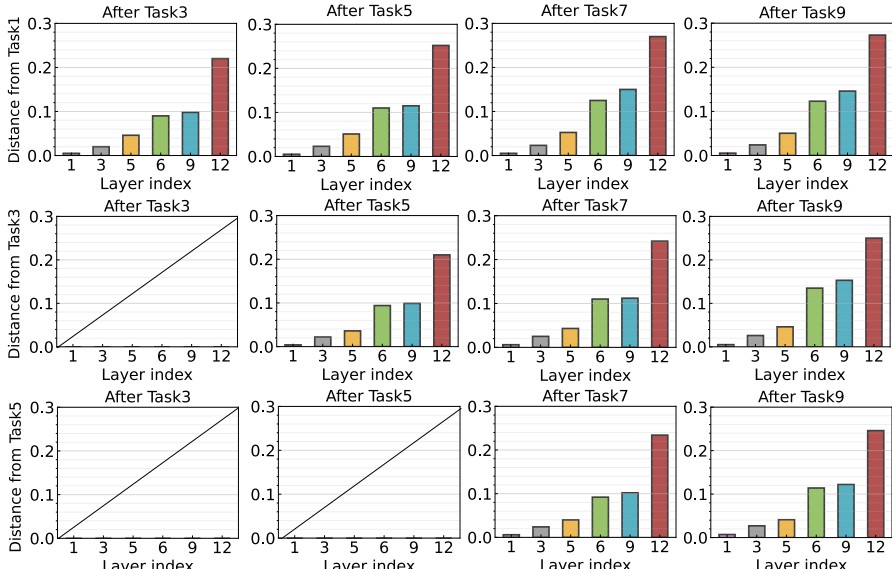

Figure 2: We measure distances between token embeddings of previous tasks when a new task is learned. Each column represents the embeddings after training a model on Tasks 3, 5, 7, and 9. Each row means the distance from the embeddings of Tasks 1, 3, and 5. For instance, the top-left figure represents the distance between token embeddings of Task 1 and those learned when Task 3 is completed. We train prompts using CodaPrompt (Smith et al., 2023a) and use $1 - CosSim(x, y)$ to measure the layer-wise distance on the training dataset. We use a CIFAR100 10-task setting.

To address the concern, we validate the deviation of the token embedding as prompt continual learning progresses. In Fig. 2, we show how this change happens at each layer during continuous learning tasks. From our study, two main observations can be highlighted: (1) When we add prompts, there is a larger difference in the deeper layers. For example, layers $1 \sim 5$ have a small difference ($\leq$ 0.1). However, the last layer shows a bigger change ($\geq 0.1$). (2) As learning continues, the earlier layers remain relatively stable, but the deeper layers change more. This observation concludes that although prompt tokens are included, the token embedding of early layers shows minor changes during training. Therefore, using token embeddings from these earlier layers would give a stable and consistent representation throughout the continual learning stages.

## 4.2 ONE-STAGE PCL FRAMEWORK

Building on these observations, we employ the token embedding from the early layers as a prompt query to generate layer-wise prompt tokens. For a fair comparison, we implement prompts across layers 1 to 5, in line with prior work. The proposed OS-Prompt framework is illustrated in Fig. 3. For each layer, given the input token embedding, we directly use the $[CLS]$ token as the prompt query. The original query selection equation (Eq. 1) can be reformulated as:

$$q_l = x_{l_{[CLS]}}. \tag{6}$$

Using the provided query $q_l$, we generate a prompt from the prompt pool following the state-of-the-art CodaPrompt (Smith et al., 2023a). It is worth highlighting that our primary contribution lies not in introducing a new prompt generation technique but in presenting a more efficient framework for query selection. We first measure the cosine similarity $\gamma(\cdot)$ between $q_l$ and keys $\{k_l^1, ..., k_l^M\}$, and then we perform a weighted summation of the corresponding value $p_l^m$ (*i.e.*, prompt) based on the similarity.

$$\phi_l = \sum_m \gamma(q_l, k_l^m) p_l^m. \tag{7}$$

The generated prompt is then prepended to the image tokens. Notably, since we produce a prompt query without the need for an extra ViT feedforward, the computational overhead is reduced by approximately $50\%$ for both training and inference.

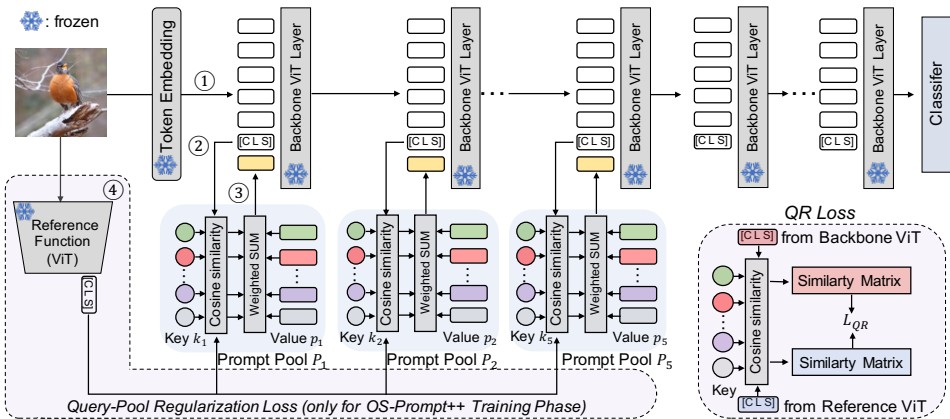

Figure 3: Our OS-Prompt (OS-Prompt++) framework. An image passes through the backbone ViT layers to get the final prediction. From layer 1 to layer 5, we prepend prompt tokens to the image tokens, which can be obtained in the following progress: ① We first compute the image token embedding from the previous layer. ② We use $[CLS]$ token as a prompt query used for measuring cosine similarity between prompt keys inside the prompt pool. ③ Based on the similarity, we do weighted sum values to obtain prompt tokens. ④ To further improve the accuracy, we present OS-Prompt++ . We integrate the query-pool regularization loss (dotted line), enabling the prompt pool to capture a stronger representation from the reference ViT.

## 4.3 QUERY-POOL REGULARIZATION LOSS

Our OS-Prompt relies on intermediate token embeddings. As a result, the prompt query exhibits diminished representational capacity compared to previous PCL approaches that utilize the $[CLS]$ token from the final layer. This reduced capacity for representation brings performance degradation. To mitigate this, we introduce a Query-Pool Regularization (QR) loss. The QR loss ensures that the query-pool relationship becomes similar to that of the final layer's $[CLS]$ token, thereby improving representation power. We extract the $[CLS]$ token from the last layer of a reference ViT architecture $R(\cdot)$ and employ it as our reference prompt query $r \in \mathbb{R}^{1 \times D}$ where $D$ is the feature dimension:

$$r = R(x). \tag{8}$$

Let $K_l \in \mathbb{R}^{M \times D}$ be the matrix representation of prompt keys at layer $l$. We then define a similarity matrix $A_{query}^l \in \mathbb{R}^{M \times 1}$ to capture the relationship between query $q_l \in \mathbb{R}^{1 \times D}$ (from Eq. 6) and prompt keys. Similarly, we compute the similarity matrix $A_{ref}^l$ to represent the relationship between the reference prompt query $r$ and the prompt keys.

$$A_{query}^l = Softmax(\frac{K_l q_l^T}{||K_l||_2 ||q_l||_2}) \qquad A_{ref}^l = Softmax(\frac{K_l r^T}{||K_l||_2 ||r||_2}). \tag{9}$$

We apply $Softmax()$ to measure the relative similarity among query-key pairs, considering that $q$ and $r$ show distinct feature distributions originating from different layers. The QR loss penalizes the distance between two similarity matrices.

$$\mathcal{L}_{QR} = \sum_l ||A_{query}^l - A_{ref}^l||_2^2. \tag{10}$$

The QR loss is added to the cross-entropy loss $\mathcal{L}_{CE}$ for classification. The total loss function can be written as:

$$\mathcal{L}_{total} = \mathcal{L}_{CE} + \lambda \mathcal{L}_{QR}, \tag{11}$$

where $\lambda$ is a hyperparameter to balance between two loss terms. Importantly, the QR loss is applied during the training phase, ensuring that there is no computational overhead during inference. Note that we only train prompts within a prompt pool while freezing the other weight parameters.

As the QR loss is an architecture-agnostic design with respect to the reference ViT (it just forwards an image and get $[CLS]$ token embedding), we leverage advanced ViT models to improve the impact of QR loss. We explore recent state-of-the-art ViT models like Swin-Transformer-v2 (Liu et al., 2022), PoolFormer (Yu et al., 2022a), and CaFormer (Yu et al., 2022a). In our experiments, we observe using a strong reference ViT improves the performance (Section 5.3).

Table 1: Results on ImageNet-R for different task configurations: 5 tasks (40 classes/task), 10 tasks (20 classes/task), and 20 tasks (10 classes/task). $A_N$ represents the average accuracy across tasks, and $F_N$ indicates the mean forgetting rate. ↑ and ↓ indicate whether a metric is better with a higher or lower value, respectively. We average values over five runs with different seeds.

| Setting | Task-5 | | Task-10 | | Task-20 | |
|---|---|---|---|---|---|---|
| Method | $A_N(\uparrow)$ | $F_N(\downarrow)$ | $A_N(\uparrow)$ | $F_N(\downarrow)$ | $A_N(\uparrow)$ | $F_N(\downarrow)$ |
| UB | 77.13 | - | 77.13 | - | 77.13 | - |
| FT | 18.74 ± 0.44 | 41.49 ± 0.52 | 10.12 ± 0.51 | 25.69 ± 0.23 | 4.75 ± 0.40 | 16.34 ± 0.19 |
| ER | 71.72 ± 0.71 | 13.70 ± 0.26 | 64.43 ± 1.16 | 10.30 ± 0.05 | 52.43 ± 0.87 | 7.70 ± 0.13 |
| LwF | 74.56 ± 0.59 | 4.98 ± 0.37 | 66.73 ± 1.25 | 3.52 ± 0.39 | 54.05 ± 2.66 | 2.86 ± 0.26 |
| L2P | 70.83 ± 0.58 | 3.36 ± 0.18 | 69.29 ± 0.73 | 2.03 ± 0.19 | 65.89 ± 1.30 | 1.24 ± 0.14 |
| Deep L2P | 73.93 ± 0.37 | 2.69 ± 0.10 | 71.66 ± 0.64 | 1.78 ± 0.16 | 68.42 ± 1.20 | 1.12 ± 0.13 |
| DualPrompt | 73.05 ± 0.50 | 2.64 ± 0.17 | 71.32 ± 0.62 | 1.71 ± 0.24 | 67.87 ± 1.39 | 1.07 ± 0.14 |
| CodaPrompt | 76.51 ± 0.38 | 2.99 ± 0.19 | 75.45 ± 0.56 | 1.64 ± 0.10 | 72.37 ± 1.19 | 0.96 ± 0.15 |
| OS-Prompt | 75.74 ± 0.58 | 3.32 ± 0.31 | 74.58 ± 0.56 | 1.92 ± 0.15 | 72.00 ± 0.60 | 1.09 ± 0.11 |
| OS-Prompt++ | **77.07 ± 0.15** | **2.23 ± 0.18** | **75.67 ± 0.40** | **1.27 ± 0.10** | **73.77 ± 0.19** | **0.79 ± 0.07** |

# 5 EXPERIMENTS

## 5.1 EXPERIMENT SETTING

**Dataset.** We utilize the Split CIFAR-100 (Krizhevsky et al., 2009) and Split ImageNet-R (Hendrycks et al., 2020) benchmarks for class-incremental continual learning. The Split CIFAR-100 partitions the original CIFAR-100 dataset into 10 distinct tasks, each comprising 10 classes. The Split ImageNet-R benchmark is an adaptation of ImageNet-R, encompassing diverse styles, including cartoon, graffiti, and origami. For our experiments, we segment ImageNet-R into 5, 10, or 20 distinct class groupings. Given its substantial intra-class diversity, the ImageNet-R benchmark is viewed as a particularly challenging benchmark. We also provide experiments on DomainNet (Peng et al., 2019), a large-scale domain adaptation dataset. We use 5 different domain as a continual learning tasks where each task consists of 69 classes.

**Experimental Details.** We performed our experiments using the ViT-B/16 (Dosovitskiy et al., 2021) pre-trained on ImageNet-1k, a standard backbone in earlier PCL research. To ensure a fair comparison with prior work, we maintain the same prompt length (8) and number of prompt components (100) as used in CodaPrompt. Like CodaPrompt, we divide the total prompt components into the number of tasks, and provide the partial component for each task. In task $N$, the prompt components from task $1 \sim N$ is frozen, and we only train the key and prompt components from task $N$. We split 20% of the training set for hyperparameter $\lambda$ tuning in Eq. 11. Our experimental setup is based on the PyTorch. The experiments utilize four Nvidia RTX2080ti GPUs. For robustness, we conducted our benchmarks with five different permutations of task class order, presenting both the average and standard deviation of results. Detailed information can be found in the Appendix B.

**Evaluation Metrics.** We use two metrics for evaluation: (1) Average final accuracy $A_N$, which measures the overall accuracy across N tasks. (Wang et al., 2022d;c; Smith et al., 2023a) (2) Average forgetting $F_N$, which tracks local performance drop over N tasks (Smith et al., 2023b;a; Lopez-Paz & Ranzato, 2017). Note, $A_N$ is mainly used for performance comparison in literature.

## 5.2 COMPARISON WITH PRIOR PCL WORKS

We compare our OS-Prompt with prior continual learning methods. This includes non-PCL methods such as ER (Chaudhry et al., 2019) and LWF (Li & Hoiem, 2017), as well as PCL methods like L2P (Wang et al., 2022d), DualPrompt (Wang et al., 2022c), and CodaPrompt (Smith et al., 2023a). We present both the upper bound (UB) performance and results from fine-tuning (FT). UB refers to training a model with standard supervised training using all data (no continual learning). FT indicates that a model undergoes sequential training with continual learning data without incorporating prompt learning. Moreover, we report L2P/Deep L2P implementation from CodaPrompt (Smith et al., 2023a), both in its original form and improved version by applying prompt pool through layers 1 to 5. In our comparisons, *OS-Prompt* represents our one-shot prompt framework without the QR loss, while *OS-Prompt++* is with the QR loss.

Table 1 shows the results on ImageNet-R. Our OS-Prompt indicates only a slight performance drop ($\leq 1\%$) across the 5-task, 10-task, and 20-task settings. The reason for the performance drop could be the reduced representational capacity of a prompt query from the intermediate token embedding.

Table 2: Accuracy comparison on CIFAR-100 10-task setting.

| Method | $A_N(\uparrow)$ | $F_N(\downarrow)$ |
|---|---|---|
| UB | 89.30 | - |
| ER | 76.20 ± 1.04 | 8.50 ± 0.37 |
| Deep L2P | 84.30 ± 1.03 | 1.53 ± 0.40 |
| DualPrompt | 83.05 ± 1.16 | 1.72 ± 0.40 |
| CodaPrompt | 86.25 ± 0.74 | 1.67 ± 0.26 |
| OS-Prompt | 86.42 ± 0.61 | 1.64 ± 0.14 |
| OS-Prompt++ | **86.68 ± 0.67** | **1.18 ± 0.21** |

Table 3: GFLOPs comparison of PCL works. We also provide relative cost (%) with respect to L2P.

| Method | Training GFLOPs | Inference GFLOPs |
|---|---|---|
| L2P | 52.8 (100%) | 35.1 (100%) |
| Deep L2P | 52.8 (100%) | 35.1 (100%) |
| DualPrompt | 52.8 (100%) | 35.1 (100%) |
| CodaPrompt | 52.8 (100%) | 35.1 (100%) |
| OS-Prompt | **35.4 (66.7%)** | **17.6 (50.1%)** |
| OS-Prompt++ | 52.8 (100%) | **17.6 (50.1%)** |

Table 4: Analysis on the impact of Reference ViT architecture on ImageNet-R.

| Setting | Task-5 | | Task-10 | | Task-20 | |
|---|---|---|---|---|---|---|
| Method (ViT Architecture) | $A_N(\uparrow)$ | $F_N(\downarrow)$ | $A_N(\uparrow)$ | $F_N(\downarrow)$ | $A_N(\uparrow)$ | $F_N(\downarrow)$ |
| OS-Prompt++ (ViT-B/16) | 77.07 ± 0.15 | 2.23 ± 0.18 | 75.67 ± 0.40 | 1.27 ± 0.10 | 73.77 ± 0.19 | 0.79 ± 0.07 |
| OS-Prompt++ (Swin-v2) | 77.27 ± 0.21 | 2.24 ± 0.24 | 75.94 ± 0.28 | 1.29 ± 0.15 | 73.89 ± 0.38 | 0.72 ± 0.11 |
| OS-Prompt++ (PoolFormer) | 77.20 ± 0.50 | 2.19 ± 0.27 | 75.86 ± 0.40 | 1.42 ± 0.24 | 73.81 ± 0.64 | 0.73 ± 0.11 |
| OS-Prompt++ (CaFormer) | 77.16 ± 0.42 | 2.16 ± 0.24 | 75.91 ± 0.54 | 1.30 ± 0.12 | 73.83 ± 0.54 | 0.72 ± 0.05 |

However, OS-Prompt++ effectively counters this limitation by incorporating the QR loss. Such an observation underscores the significance of the relationship between the query and the prompt pool in PCL. OS-prompt++ shows slight performance improvement across all scenarios. This implies a prompt query that contains task information helps to enhance representation in the prompt pool, suggesting that exploring methods to integrate task information inside the prompt selection process could be an interesting research direction in PCL. Table 2 provides results on 10-task CIFAR-100, which shows a similar trend with ImageNet-R. We also provide DomainNet results in Appendix H

In Table 3, we provide GFLOPs for various PCL methods, rounding GFLOPs to one decimal place. It is worth noting that while prior PCL methods might have marginally different GFLOP values, they are close enough to appear identical in the table. This implies that different prompt formation schemes proposed in prior works do not substantially impact GFLOPs. The table shows OS-Prompt operates at 66.7% and 50.1% of the GFLOPs, during training and inference, respectively, relative to prior methods. While OS-Prompt++ maintains a training GFLOPs count comparable to earlier work due to the added feedforward process in the reference ViT, it does not employ the reference ViT during infer-

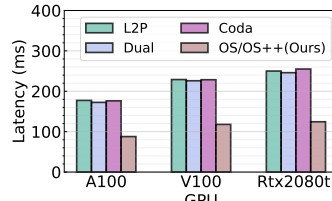

Figure 4: Comparison of latency across PCL methods on different GPU models.

ence, bringing it down to 50.1% GFLOPs. In Fig. 4, we compare the inference GPU latency of previous PCL methods with our OS-Prompt across three distinct GPU setups. All measurements are taken using a batch size of 32. Similar to the trend observed in FLOPs, our approach reduces the latency by $\sim 50\%$. We provide further discussion on training computational cost in Appendix G.

### 5.3 DOES STRONGER REFERENCE ViT IMPROVE PERFORMANCE?

The QR loss is an architecture-agnostic design with respect to the reference ViT because it only requires forwarding an image to obtain the $[CLS]$ token embedding. To amplify the effects of the QR loss, we experiment with cutting-edge ViT models such as Swin-Transformer-v2 (Liu et al., 2022), PoolFormer (Yu et al., 2022a), and CaFormer (Yu et al., 2022a). For a fair comparison, we ensure that all these advanced architectures are trained on ImageNet-1k. As presented in Table 4, leveraging advanced reference ViT architectures improves accuracy by providing stronger representations. For example, compared to ViT-B/16 baseline, using Swin-v2 improves the $A_N$ metric by 0.2%/0.27%/0.12% on Task-5/Task-10/Task-20, respectively. This result underscores the advantage of our method, emphasizing its adaptability with future advanced ViT architectures. We also explore the advantage of applying the advanced ViT architectures to the prior work in Appendix D.

### 5.4 ANALYSIS OF DESIGN COMPONENTS

**QR Loss Design.** To understand the effect of different components in our proposed QR loss (Eq. 10), we conducted an ablation study with different settings, summarized in Table 5. There are two main components in the QR loss: *Cosine Similarity* and *Softmax*, and we provide the accuracy of different four combinations. Without Cosine Similarity and Softmax, our framework yields a

Table 5: Analysis of QR loss design. We train OS-Prompt++ on 10-task ImageNet-R setting.

| CosSim | Softmax | $A_N(\uparrow)$ | $F_N(\downarrow)$ |
|---|---|---|---|
| | | $75.00 \pm 0.53$ | $1.68 \pm 0.12$ |
| | ✓ | $75.47 \pm 0.42$ | $1.38 \pm 0.16$ |
| ✓ | | $75.51 \pm 0.33$ | $1.28 \pm 0.06$ |
| ✓ | ✓ | $75.67 \pm 0.40$ | $1.27 \pm 0.10$ |

Table 6: Hyperparameter sensitivity study of $\lambda$ on ImageNet-R 5/10/20-task.

| $\lambda$ | Task-5 | Task-10 | Task-20 |
|---|---|---|---|
| 1e-5 | $77.03 \pm 0.10$ | $75.63 \pm 0.39$ | $73.63 \pm 0.21$ |
| 5e-5 | $77.02 \pm 0.13$ | $75.62 \pm 0.41$ | $73.62 \pm 0.19$ |
| 1e-4 | $77.07 \pm 0.15$ | $75.67 \pm 0.40$ | $73.77 \pm 0.19$ |
| 5e-4 | $77.13 \pm 0.24$ | $75.68 \pm 0.38$ | $73.68 \pm 0.17$ |

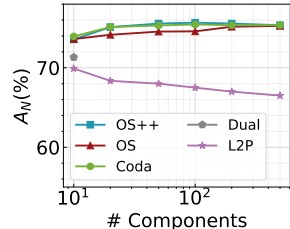 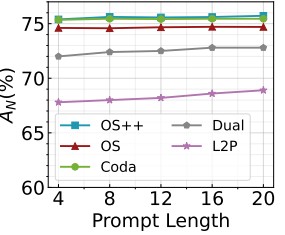 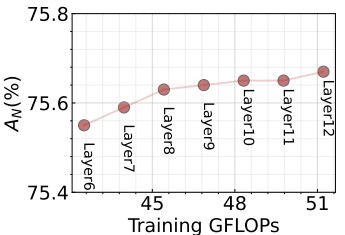

Figure 5: Analysis of accuracy $A_N$ with respect to prompt components (**Left**) and prompt length (**Right**). We use 10-taks ImageNet-R setting.

Figure 6: Trade-off between GFLOPs and accuracy within a reference ViT.

performance of $75.00\%$ for $A_N$. Adding Cosine Similarity or Softmax improves the performance, suggesting their collaborative role in enhancing the model's performance in our QR loss design. We also provide the sensitivity analysis of our method with respect to the hyperparameter $\lambda$ in Table 6. Across the three distinct task configurations on ImageNet-R, we note that the performance fluctuations are minimal, despite the variation in $\lambda$ values. These results underline the robustness of our method with respect to $\lambda$ value, suggesting that our approach is not sensitive to hyperparameter.

**Prompt Design.** We further examine the impact of both the number of prompt components and the prompt length on accuracy. In Fig. 5 (left), we measure accuracy across configurations with $\{10, 20, 50, 100, 200, 500\}$ prompts. The OS-prompt demonstrates a consistent accuracy enhancement as the number of prompts increases. Notably, OS-prompt++ reaches a performance plateau after just 50 prompts. Additionally, our ablation study of prompt length, presented in Fig. 5 (right), reveals that our method maintains stable accuracy across various prompt lengths.

## 5.5 ACCURACY-EFFICIENCY TRADE-OFF WITHIN A REFERENCE VIT

During the training of OS-prompt++, our method does not improve the energy efficiency (Note, we achieve $\sim 50\%$ computational saving during inference). This arises from our approach of extracting the reference prompt query $r$ from the final layer of the reference ViT. To further enhance energy efficiency during the OS-prompt++ training, we delve into the trade-off between accuracy and efficiency within the Reference ViT. Instead of relying on the last layer's $[CLS]$ token embedding for the reference prompt query, we opt for intermediate token embeddings. Since our prompt pool is applied in layers 1 to 5 of the backbone ViT, we utilize the intermediate token embeddings from layers deeper than 5 within the reference ViT. Fig. 6 illustrates this trade-off with respect to the layer index where we get the reference prompt. Our findings show that utill layer 8, there is only a slight increase in accuracy, which suggests a potential to reduce GFLOPs without performance drop.

## 6 CONCLUSION

In this paper, we introduce the OS-Prompt framework where we improve the efficiency of the conventional two-stage PCL in a simple yet effective manner. By harnessing the power of intermediate token embeddings for prompts and introducing the Query-Pool Regularization (QR) loss, we save computational cost without performance drop on class-incremental continual learning scenarios. One of the potential limitations of our OS-Prompt++ is, in comparison to OS-Prompt (our light version), it introduces an increase in training computational cost from the reference ViT feedforward. This computational efficiency during training becomes important in the context of addressing the online continual learning problem (Ghunaim et al., 2023) where the model needs rapid training on the given streaming data. To further improve the efficiency, we can adopt the early exit in (Phuong & Lampert, 2019), where they make predictions in the middle layers.

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

## A   CODE IMPLEMENTATION

We provide a code with **.zip** format in Supplementary Materials. Our code is based on the CodaPrompt official Github implementation [1]. The codebase contains the prior PCL methods.

## B   EXPERIMENTAL DETAILS

We maintain the prompting configuration (*i.e.*, lengths and locations) for L2P and DualPrompt as suggested by the CodaPrompt paper. The detailed configurations are as follows: L2P incorporates a prompt pool of size 20, a total prompt length of 20, and selects 5 prompts from the pool for inference. DualPrompt utilizes a length 5 prompt in layers $1 \sim 2$ and length 20 prompts in layers $3 \sim 5$. CodaPrompt uses a prompt length of 8 and 100 prompt components. For all PCL methods, including our OS-prompt, we employ the Adam optimizer (Kingma & Ba, 2014) with $\beta_1 = 0.9$ and $\beta_2 = 0.999$, using a batch size of 128 images. We use cosine-decaying learning rate scheduling with learning rate $1e-3$. All images are resized to 224x224 and normalized. CIFAR-100 is trained for 20 epochs, and ImageNet-R for 50 epochs.

To train a prompt pool in our method, we freeze prompts from the previous stages ($1 \sim N-1$) and only train $N$ stage's prompt. For example, for 10-task class incremental continual learning, we add 10 prompts to the prompt pool for each stages and only train newly added 10 prompts. To initialize the new prompts, we use Gram-Schmidt process (Pursell & Trimble, 1991) at the start of each new task, as described in CodaPrompt official Github implementation. Furthermore, in alignment with prior PCL implementations, we substitute the predictions from prior-task logits with negative infinity during the training phase of the current task. Note that we do not use the query attention proposed in CodaPrompt.

## C   ADVANCED ViT ARCHITECTURE DETAILS

Here we provide the architecture details of advanced ViTs used in our work. We use pre-trained models from PyTorch Image Models (timm) [2].

• **Swin-Transformer-v2** (Liu et al., 2022) introduces three primary technical contributions to advance large-scale models in computer vision: firstly, a combined residual-post-norm method with cosine attention to enhance training stability; secondly, a log-spaced continuous position bias technique that allows seamless transfer from low-resolution pre-training to high-resolution downstream tasks; and thirdly, SimMIM, a self-supervised pre-training method designed to curtail the reliance on vast quantities of labeled images. We use SwinV2-S model in our work.

• **PoolFormer** (Yu et al., 2022a) adopts a fundamental approach by utilizing pooling as the token mixer in their model. This pooling operator, distinguished by its lack of trainable parameters, straightforwardly averages the features of neighboring tokens. We use PoolFormer-M36 model in our work.

• **CaFormer** (Yu et al., 2022b) incorporates depthwise separable convolutions in the initial layers and employing standard self-attention in the later layers. The model establishes $85.5\%$ accuracy on ImageNet-1K without any reliance on external data sources or the use of distillation techniques. We use CaFormer-M36 model in our work.

## D   STRONGER QUERY ViT IN TWO-STAGE PCL FRAMEWORK

We investigate the benefits of integrating advanced ViT architectures into previous PCL approaches. In Table 7, we provide the performance of various ViT architectures on 10-task ImageNet-R setting.

L2P with ViT-B/16 yields an $A_N$ of 69.29, whereas employing the Swin-v2, PoolFormer, and CaFormer architectures results in a notable performance improvement with $A_N$ scores by $\sim 4.3\%$.

---

[1] https://github.com/GT-RIPL/CODA-Prompt
[2] https://huggingface.co/timm

Table 7: Analysis on the impact of ViT architecture on ImageNet-R.

| Setting | Task-10 | |
|---|---|---|
| Method (ViT Architecture) | $A_N(\uparrow)$ | $F_N(\downarrow)$ |
| L2P (ViT-B/16) | 69.29 ± 0.73 | 2.03 ± 0.19 |
| L2P (Swin-v2) | 73.53 ± 0.41 | 1.36 ± 0.15 |
| L2P (PoolFormer) | 73.60 ± 0.34 | 1.42 ± 0.16 |
| L2P (CaFormer) | 73.63 ± 0.43 | 1.40 ± 0.10 |
| DualPrompt (ViT-B/16) | 71.32 ± 0.62 | 1.71 ± 0.24 |
| DualPrompt (Swin-v2) | 72.62 ± 0.58 | 1.12 ± 0.18 |
| DualPrompt (PoolFormer) | 72.61 ± 0.47 | 1.07 ± 0.07 |
| DualPrompt (CaFormer) | 72.83 ± 0.49 | 1.24 ± 0.14 |
| CodaPrompt (ViT-B/16) | 75.45 ± 0.56 | 1.64 ± 0.10 |
| CodaPrompt (Swin-v2) | 76.16 ± 0.36 | 1.25 ± 0.17 |
| CodaPrompt (PoolFormer) | 76.12 ± 0.10 | 1.08 ± 0.13 |
| CodaPrompt (CaFormer) | 76.55 ± 0.34 | 1.05 ± 0.05 |

Table 8: Analysis on prompt formation methods. Results on ImageNet-R for different task configurations: 5 tasks (40 classes/task), 10 tasks (20 classes/task), and 20 tasks (10 classes/task). $A_N$ represents the average accuracy across tasks, and $F_N$ indicates the mean forgetting rate. We average values over five runs with different seeds.

| Setting | Task-5 | | Task-10 | | Task-20 | |
|---|---|---|---|---|---|---|
| Method | $A_N(\uparrow)$ | $F_N(\downarrow)$ | $A_N(\uparrow)$ | $F_N(\downarrow)$ | $A_N(\uparrow)$ | $F_N(\downarrow)$ |
| UB | 77.13 | - | 77.13 | - | 77.13 | - |
| Deep L2P | 73.93 ± 0.37 | 2.69 ± 0.10 | 71.66 ± 0.64 | 1.78 ± 0.16 | 68.42 ± 1.20 | 1.12 ± 0.13 |
| OS-Prompt (L2P) | 74.61 ± 0.29 | 2.29 ± 0.21 | 73.43 ± 0.60 | 1.42 ± 0.10 | 71.42 ± 0.24 | 0.95 ± 0.06 |
| DualPrompt | 73.05 ± 0.50 | 2.64 ± 0.17 | 71.32 ± 0.62 | 1.71 ± 0.24 | 67.87 ± 1.39 | 1.07 ± 0.14 |
| OS-Prompt (Dual) | 74.65 ± 0.15 | 2.06 ± 0.20 | 72.57 ± 0.23 | 1.13 ± 0.03 | 70.55 ± 0.48 | 0.81 ± 0.10 |
| CodaPrompt | 76.51 ± 0.38 | 2.99 ± 0.19 | 75.45 ± 0.56 | 1.64 ± 0.10 | 72.37 ± 1.19 | 0.96 ± 0.15 |
| OS-Prompt (Coda) | 75.74 ± 0.58 | 3.32 ± 0.31 | 74.58 ± 0.56 | 1.92 ± 0.15 | 72.00 ± 0.60 | 1.09 ± 0.11 |

For the DualPrompt method, the ViT-B/16 architecture provides an $A_N$ of 71.32%. Transitioning to more advanced architectures improves the performance by $\sim 1.5\%$. Similarly, CodaPrompt shows performance improvement with Swin-v2, PoolFormer, and CaFormer architectures compared to ViT-B/16 architecture. In sum, the results suggest that regardless of the chosen method, more advanced architectures such as Swin-v2, PoolFormer, and CaFormer tend to outperform the ViT-B/16 configuration. Nevertheless, incorporating a more advanced query ViT does not alleviate computational cost; in fact, it may substantially increase computational cost. Conversely, while our OS-Prompt++ technique utilizes the strengths of advanced ViT architectures, we do not need to deploy the advanced ViT during inference, resulting in energy-efficiency.

## E  ANALYSIS ON PROMPT FORMATION STRATEGY

In our approach, the construction of prompts is achieved through a weighted summation of components within the prompt pool, mirroring the strategy employed by CodaPrompt. In this section, we delve into the impact of varying prompt formation strategies on performance. For comparative insight, we present results from two previous PCL methodologies: L2P and DualPrompt. While L2P selects 5 prompts from the internal prompt pool, DualPrompt distinguishes between general and task-specific prompts. Aside from the distinct prompt formation strategies, the configurations remain consistent.

In Table 8, we adopt the prompt formation strategies proposed in L2P and Dual, resulting in configurations denoted as OS-Prompt (L2P) and OS-Prompt (Dual). Our findings reveal the following: (1) The effectiveness of OS-prompt varies based on the prompt formation technique employed. Notably, OS-Prompt (L2P) and OS-Prompt (Dual) yield lower accuracy than our original approach, which relies on CodaPrompt. This suggests that our method's performance could benefit from refined prompt formation techniques in future iterations. (2) The OS-prompt framework, when integrated with L2P and Dual prompt formation strategies, outperforms the original Deep L2P and DualPrompt, a trend

not observed with CodaPrompt. This may indicate that the prompt formation strategies of L2P and DualPrompt, which rely on hard matching (i.e., top-k), are more resilient than CodaPrompt's. Conversely, CodaPrompt employs soft matching, utilizing a weighted summation of all prompts based on proximity. This might be more susceptible to the diminished representation power from the intermediate layer's features. Simultaneously, a prompt query imbued with task-specific data (since our query token incorporates task prompts) appears to enhance representation within the prompt pool.

## F ALGORITHM

In this study, we introduce the algorithms of both the prior PCL and our OS-Prompt, detailed in Algorithm 1 and 2, respectively. As discussed in the primary text, earlier PCL methods deploy a query ViT to produce a prompt query, as seen in line 2 of Algorithm 1. With this generated prompt query, prompts are formed during the feedforward process, as illustrated in lines 3 to 5 of Algorithm 1. In contrast, our approach utilizes the intermediate [CLS] token as a prompt query, as indicated in line 4 of Algorithm 2.

---

**Algorithm 1** Prior PCL works

---

Input image $x$, Query ViT $Q(\cdot)$, BackBone ViT $F(\cdot)$ consists of layer $f_l$, where $l \in \{1, ..., L\}$, prompt formation function $g()$, Prompt pool $P_l$

1: # feedforward step with image $x$
2: $q = Q(x)$          ▷ Compute a prompt query
3: **for** $l \leftarrow 1$ to 5 **do**          ▷ Apply a prompt pool for the first five layers
4:      $\phi_l \leftarrow g(q, P_l)$
5:      $x_{l+1} \leftarrow f_l(x_l, \phi_l)$
6: **end for**
7: **for** $l \leftarrow 6$ to $L$ **do**          ▷ Standard ViT operation for leftover layers
8:      $x_{l+1} \leftarrow f_l(x_l)$
9: **end for**
10: **return** $Classifier(x_{L+1})$

---

---

**Algorithm 2** OP-Prompt (Ours)

---

Input image $x$, BackBone ViT $F(\cdot)$ consists of layer $f_l$, where $l \in \{1, ..., L\}$, prompt formation function $g(\cdot)$, Prompt pool $P_l$

1: # feedforward step with image $x$
2: **for** $l \leftarrow 1$ to 5 **do**          ▷ Apply a prompt pool for the first five layers
3:      $q_l \leftarrow x_{l_{[CLS]}}$          ▷ Use [CLS] token as a prompt query
4:      $\phi_l \leftarrow g(q_l, P_l)$
5:      $x_{l+1} \leftarrow f_l(x_l, \phi_l)$
6: **end for**
7: **for** $l \leftarrow 6$ to $L$ **do**          ▷ Standard ViT operation for leftover layers
8:      $x_{l+1} \leftarrow f_l(x_l)$
9: **end for**
10: **return** $Classifier(x_{L+1})$

---

## G DISCUSSION ON TRAINING COMPUTATIONAL COST

In this section, we provide an in-depth discussion of the training computational cost of PCL methods. Following (Ghunaim et al., 2023), Table 9 presents the relative training complexity of each method with respect to ER (Chaudhry et al., 2019), which is the simple baseline with standard gradient-based training.

Here, we would like to clarify the training computational cost of prompt learning. In general neural network training, the forward-backward computational cost ratio is approximately 1:2. This is due to gradient backpropagation ($\frac{dL}{da_l} = W_{l+1} \frac{dL}{da_{l+1}}$) and weight-updating ($\frac{dL}{dW_l} = \frac{dL}{da_{l+1}} a_l$), where $L$ represents the loss, $W$ denotes the weight parameter, $a$ is the activation, and $l$ is the layer index.

Table 9: Relative training computational cost.

| CL strategy | Method | Relative training complexity w.r.t ER |
|---|---|---|
| Replay-based | ER | 1 |
| Regularization-based | LWF | 4/3 |
| Prompt-based | L2P | 1 |
| Prompt-based | DualPrompt | 1 |
| Prompt-based | CodaPrompt | 1 |
| Prompt-based | OS-Prompt (Ours) | 2/3 |
| Prompt-based | OS-Prompt++ (Ours) | 1 |

In prompt learning, the forward-backward computational cost ratio is approximately 1:1. This is in contrast to general neural network training, as only a small fraction of the weight parameters (less than 1%) are updated.

Bearing this in mind, we present the observations derived from the table, assuming that all methods employ the same architecture.
• Previous PCL (L2P, DualPrompt, CodaPrompt) consists of two steps; First, the query ViT requires only feedforward without backpropagation; Second, the backbone ViT feedforward-backward training with prompt tuning. This results in the relative training computational cost is 1 $(= \frac{QueryViTforward(1)+BackboneViTforward(1)+BackboneViTbackward(1)}{ERforward(1)+ERbackward(2)})$.
• Similarly, our OS-Prompt++ has a reference ViT which only requires a feedforward step, bringing 1 relative training computational cost with respect to ER.
• On the other hand, our OS-Prompt requires one feedforward-backward step like standard training. Therefore, relative training computaional cost becomes $\frac{2}{3}$ $(= \frac{BackboneViTforward(1)+BackboneViTbackward(1)}{ERforward(1)+ERbackward(2)})$. This shows the potential of our OS-Prompt on online continual learning.

Overall, we propose two versions of the one-stage PCL method (i.e., OS-prompt and OS-prompt++), and these two options enable the users can select a suitable method depending on the problem setting. For example, for online continual learning, OS-prompt is a better option since it requires less training cost. On the other hand, for offline continual learning, one can use OS-prompt++ to maximize the performance while spending more training energy.

## H DOMAINNET EXPERIMENTS

We conducted experiments on DomainNet (Peng et al., 2019), a large-scale domain adaptation dataset consisting of around 0.6 million images distributed across 345 classes. Each domain involves 40,000 images for training and 8,000 images for testing. Our continual domain adaptation task was created using five diverse domains from DomainNet: Clipart → Real → Infograph → Sketch → Painting. The experimental methodology closely followed the protocol established in prior PCL research (Smith et al., 2023a). The outcomes of these experiments are presented in Table 10. Our method (OS-prompt and OS-propmt++) achieves comparable performance compared to the previous PCL methods with 50% inference computational cost. These results affirm the practical viability of our approach, indicating its applicability in real-world scenarios.

## I IMPACT OF UNSUPERVISED PRE-TRAINED WEIGHTS

One underlying assumption of existing PCL methods is their reliance on supervised pretraining on ImageNet-1k. While this pre-training has a substantial impact on model performance, it may not always be feasible for a general continual learning task. One possible solution is using unsupervised pre-trained models. To explore this, we use DINO pre-trained weights (Caron et al., 2021) instead of ImageNet-1k pre-trained weights, and compare the performance across ER, LwF, and PCL works. We set the other experimental settings as identical.

Table 10: Performance comparison on DomainNet 5-task setting.

| Method | $A_N(\uparrow)$ | $F_N(\downarrow)$ |
|---|---|---|
| UB | 79.65 | - |
| FT | $18.00 \pm 0.26$ | $43.55 \pm 0.27$ |
| ER | $58.32 \pm 0.47$ | $26.25 \pm 0.24$ |
| L2P | $69.58 \pm 0.39$ | $2.25 \pm 0.08$ |
| Deep L2P | $69.58 \pm 0.39$ | $2.25 \pm 0.08$ |
| DualPrompt | $70.73 \pm 0.49$ | $2.03 \pm 0.22$ |
| CodaPrompt | $73.24 \pm 0.59$ | $3.46 \pm 0.09$ |
| OS-Prompt | $72.24 \pm 0.13$ | $2.94 \pm 0.02$ |
| OS-Prompt++ | $73.32 \pm 0.32$ | $2.07 \pm 0.06$ |

We report the results in Table 11. The non-PCL methods (ER and LwF) demonstrate similar or even superior performance compared to DualPropmt and L2P. This contrasts with the results obtained using the ImageNet-1k pretrained model, where PCL methods outperformed non-PCL methods. This observation suggests that the backbone model plays a crucial role in PCL. However, CodaPrompt and our OS-Prompt outperform the other methods by a significant margin. This indicates that our method continues to perform well even without utilizing supervised pretraining.

Table 11: Analysis on the impact of unsupervised Pre-trained Weights. We use DINO pre-trained weights (Caron et al., 2021).

| Method | $A_N(\uparrow)$ | $F_N(\downarrow)$ |
|---|---|---|
| ER | $60.43 \pm 1.16$ | $13.30 \pm 0.12$ |
| LwF | $62.73 \pm 1.13$ | $4.32 \pm 0.63$ |
| L2P | $60.32 \pm 0.56$ | $2.30 \pm 0.11$ |
| DualPrompt | $61.77 \pm 0.61$ | $2.31 \pm 0.23$ |
| CodaPrompt | $67.61 \pm 0.19$ | $2.23 \pm 0.29$ |
| OS-Prompt | $67.52 \pm 0.34$ | $2.32 \pm 0.13$ |
| OS-Prompt++ | $67.92 \pm 0.42$ | $2.19 \pm 0.26$ |

