# OpenReview forum: "One-stage Prompt-based Continual Learning"
_ICLR.cc/2024/Conference — Submitted to ICLR 2024_

### Official Review · Reviewer_2Phe · 2023-11-01

**Soundness:** 3 good
**Presentation:** 3 good
**Contribution:** 3 good
**Rating:** 6
**Confidence:** 4

**Summary:**

The paper presents a one-stage prompt-based continual learning strategy that simplifies the query design and improves computational efficiency. In particular, it directly uses the intermediate layer's token embedding as a prompt query and introduces a query-pool regularization strategy to enhance the representation power. The experimental evaluation shows that the proposed method achieves about 50\% computation cost reduction and better performance on two benchmarks.

**Strengths:**

- The paper is well-written and easy to follow.

- The proposed idea is well-motivated and the method is simple and effective.

**Weaknesses:**

- Assumption on the pretrained network. One limitation of existing PCL methods is that they rely on supervised pretraining on ImageNet1k, which has a large impact on the model performance but can be infeasible for a general CL task. Does this method still work without using supervised pretraining? For instance, what if replacing the supervised pretraining model with an unsupervised pretraining model (e.g. DINO [1])?

[1] Caron et al. Emerging Properties in Self-Supervised Vision Transformers. ICCV2021.

- Lack of clarity in experimental results. In particular, Figure 2 is not very clear. Is the token embedding distance depicted in this Figure related to the current task? What are the distances between token embeddings for tasks 1 to t-1 when task t is learned? Such an illustration could provide more insights into how the method alleviates catastrophic forgetting.

**Questions:**

See the comments in the Weaknesses section.

---

> ### Author Response · Authors · 2023-11-16
> **Rebuttal**
>
> We appreciate your positive feedback on our work. Please check our response to your questions and concerns.
>
> Q1: Does this method still work without using supervised pretraining? For instance, what if replacing the supervised pretraining model with an unsupervised pretraining model (e.g. DINO [1])?
>
> A1: Thank you for the valuable feedback. We think the reviewer's suggestion is interesting and important for PCL works. In response to this suggestion, we use DINO pre-trained weights instead of ImageNet-1k pre-trained weights, and compare the performance across ER, LwF, and PCL works. We set the other experimental settings as identical.
>
> | Method (DINO pre-trained method) |    Acc (%)    | Forgetting Score |
> |----------------------------------|:-------------:|:----------------:|
> | ER                               |  60.43 ± 1.16 |   13.30 ± 0.12   |
> | LwF                              | 62.73 ± 1.13  |    4.32 ± 0.63   |
> | L2P                              |  60.32 ± 0.56 |    2.30 ± 0.11   |
> | DualPrompt                       |  61.77 ± 0.61 |    2.31 ± 0.23   |
> | CodaPrompt                       |  67.61 ± 0.19 |    2.23 ± 0.29   |
> | OS-Prompt                        |  67.52 ± 0.34 |    2.32 ± 0.13   |
> | OS-Prompt++                      |  67.92 ± 0.42 |    2.19 ± 0.26   |
>
> The non-PCL methods (ER and LwF) demonstrate similar or even superior performance compared to DualPropmt and L2P. This contrasts with the results obtained using the ImageNet-1k pre-trained model, where PCL methods outperformed non-PCL methods. This observation suggests that the backbone model plays a crucial role in PCL. However, CodaPrompt and our OS-Prompt outperform the other methods by a significant margin. This indicates that our method continues to perform well even without utilizing supervised pertaining.
>
> We have included the experiments and analysis related to the above observations in the Appendix.
>
> Q2: Lack of clarity in experimental results. In particular, Figure 2 is not very clear. Is the token embedding distance depicted in this Figure related to the current task? What are the distances between token embeddings for tasks 1 to t-1 when task t is learned?
>
> A2: Thanks for the insightful suggestion. In response to the reviewer's feedback, we have restructured Fig. 2 in our manuscript. This figure depicts the measurement of distances between token embeddings of previous tasks when a new task is learned. We conducted experiments on 9 pairs, assessing the distance between token embeddings of task 1 and those after tasks {3, 5, 7, 9}; the distance between token embeddings of tasks 3 and those after tasks {5, 7, 9}; and the distance between token embeddings of tasks 5 and those after tasks {7, 9}. The results reveal that token embeddings from early layers show minimal changes during training. Consequently, utilizing token embeddings from these earlier layers (layers  1~5) provides a stable and consistent representation throughout the continual learning stages.

---

> > ### Comment · Reviewer_2Phe · 2023-11-23
> > **Thanks for your response**
> >
> > Thanks for providing additional analysis and clarification. The results demonstrate the efficacy of the proposed method on the main benchmark. On the other hand, as the algorithms show different properties, it seems that a more comprehensive evaluation should be included for all the datasets.  Overall, I would maintain my original rating.

---

### Official Review · Reviewer_Wfb2 · 2023-11-03

**Soundness:** 3 good
**Presentation:** 2 fair
**Contribution:** 2 fair
**Rating:** 5
**Confidence:** 4

**Summary:**

This paper introduces the OS-Prompt framework, a new one-stage Prompt-Based Continual Learning (PCL) approach for image recognition tasks. By restructuring the traditional two-step feedforward stages, the OS-Prompt aims to optimize computational costs in PCL. Experiments are conducted on datasets such as Split CIFAR-100 and Split ImageNet-R, comparing the efficacy of OS-Prompt with other continual learning methodologies.

**Strengths:**

1.	Addressing Computational Inefficiencies: The paper astutely identifies a significant limitation in current PCL methods, notably the high computational cost stemming from the two separate ViT feed-forward stages. The introduction of the OS-Prompt as a remedy is commendable, achieving an impressive reduction in computational cost by nearly 50% without compromising on performance.
2.	Innovative QR Loss Introduction: In response to the minor performance decline observed with the one-stage PCL framework, the paper introduces a QR loss. This strategic addition ensures that the prompt pool remains consistent with token embedding from both intermediate and final layers. Importantly, the implementation of the QR loss is efficient, adding no extra computational burden during the inference phase.

**Weaknesses:**

1.	Potential Scalability Concerns: While the OS-Prompt framework demonstrates efficiency improvements on benchmarks like CIFAR-100 and ImageNet-R, the paper does not provide insights into how this method scales with larger, more complex datasets. This leaves questions about its applicability in broader, real-world scenarios where data variability and volume might be significantly higher.
2.	Lack of Exploration on QR Loss Limitations: The introduction of the QR loss is innovative, but the paper could benefit from a more in-depth discussion on its potential limitations or scenarios where it might not be as effective. A deeper dive into the trade-offs associated with the QR loss would provide a more balanced view of its utility.
3.	Comparative Analysis Depth: While the paper highlights the superiority of the OS-Prompt over the CodaPrompt method, it might be beneficial to see how the proposed framework fares against a wider array of contemporary methods. A more extensive comparative analysis would offer readers a comprehensive understanding of where OS-Prompt stands in the broader landscape of PCL techniques.

**Questions:**

Please the weakness.

---

> ### Author Response · Authors · 2023-11-16
> **Rebuttal**
>
> Thank you for your efforts in reviewing our article and providing constructive feedback. We’d like to reply to your concerns in detail.
>
> Q1: Potential Scalability Concerns. The paper does not provide insights into how this method scales with larger, more complex datasets.
>
> A1: We appreciate the insightful suggestion from the reviewer. In response to their input, we conducted experiments on DomainNet [R1], a large-scale domain adaptation dataset consisting of around 0.6 million images distributed across 345 categories. Each domain involves 40,000 images for training and 8,000 images for testing. Our continual domain adaptation task was created using five diverse domains from DomainNet: Clipart → Real → Infograph → Sketch → Painting. The experimental methodology closely followed the protocol established in prior PCL research. The outcomes of these experiments are presented in the table below.
>
> | Method (Dataset: DomainNet) |    Acc (%)   | Forgetting score |
> |-----------------------------|:------------:|:----------------:|
> | UppderBound                 |     79.65    |         -        |
> | FT                          | 18.00 ± 0.26 |   43.55 ± 0.27   |
> | ER                          | 58.32 ± 0.47 |   26.25 ± 0.24   |
> | L2P                         | 69.58 ± 0.39 |    2.25 ± 0.08   |
> | Deep L2P                    | 70.54 ± 0.51 |    2.05 ± 0.07   |
> | DualPrompt                  | 70.73 ± 0.49 |    2.03 ± 0.22   |
> | CODA                        | 73.24 ± 0.59 |    3.46 ± 0.09   |
> | OS-Prompt                   | 72.24 ± 0.13 |    2.94 ± 0.02   |
> | OS-Prompt++                 | 73.32 ± 0.32 |    2.07 ± 0.06   |
>
> Our method (OS-prompt and OS-propmt++) achieves comparable performance compared to the previous PCL methods with ~50% inference computational cost. These results affirm the practical viability of our approach, indicating its applicability in real-world scenarios. We have added the results in the Appendix.
>
> [R1] Peng et al. Moment matching for multi-source domain adaptation. ICCV19
>
> Q2: Lack of Exploration on QR Loss Limitations. A deeper dive into the trade-offs associated with the QR loss would provide a more balanced view of its utility.
>
> A2: Thank you for your suggestion. During the training of OS-Prompt++, in comparison to OS-Prompt, the inclusion of QR loss does enhance accuracy. However, QR loss requires a reference ViT feedforward step (which is different ViT from the backbone ViT), resulting in additional training computational cost compared to OS-Prompt. It's worth highlighting that while OS-Prompt++ has a high training computational cost compared to OS-Prompt, it remains consistent with the training cost of prior PCL works (refer to Table 3).
>
> This computational efficiency during training becomes important in the context of addressing the online continual learning problem [R2]. This problem assumes the model needs rapid training on the given streaming data. If the training cost requires a high computational cost, the stream does not wait for the model to complete training before revealing the next data for predictions, leading to performance degradation. Therefore, an analysis of the training efficiency of OS-Prompt++ is required.
>
> To address this issue, in Fig. 6, we provide the trade-off between FLOPs and accuracy within a reference ViT used for QR Loss. In our OS-Prompt++ default setting, we extract the reference prompt query r from the final layer of the reference ViT. This requires additional training complexity compared to OS-Prompt. To improve efficiency, instead of using the final layer’s [CLS] token embedding for prompt query r, we utilize the intermediate token embeddings from layers deeper than 5 within the reference ViT. The results show there is an accuracy and training efficiency trade-off in our QR Loss design.
>
> More importantly, as per the reviewer's suggestion, we have added content in the Conclusion section to point out the potential limitations of the proposed method.
>
> [R2] Real-Time Evaluation in Online Continual Learning: A New Hope, Y. Ghunaim et al., CVPR2023

---

> > ### Author Response · Authors · 2023-11-16
> > **Rebuttal (Continue)**
> >
> > Q3: Comparative Analysis Depth. A more extensive comparative analysis would offer readers a comprehensive understanding of where OS-Prompt stands in the broader landscape of PCL technique.
> >
> > A3: Thank you for the suggestions. Within the domain of PCL techniques, our primary objective is to improve energy efficiency, setting our approach complementary to previous PCL work that primarily targeted improving the accuracy of rehearsal-free continual learning.
> >
> > While our comparison focuses on the state-of-the-art PCL method (CodaPrompt), we provide a comprehensive comparison/analysis across our method and other PCL techniques. The details are outlined below:
> >
> > - We present accuracy and relative computational cost comparisons in Tables 1, 2, and 3, with a summarized overview depicted in Figure 1 in the introduction.
> > - [Figure 5] In addition to CodaPrompt, we have included data points for L2P and DualPrompt to further enrich the comparative analysis across these PCL methods.
> > - [Table 8 in Appendix] Since our goal is orthogonal to prior work, we apply our strategy to previous PCL methods, specifically L2P and DualPrompt. This shows that our one-stage PCL can be applied to future methods incorporating advanced prompt formation strategies.
> >
> > Please let us know if you think further specific experiments are required.

---

> > > ### Comment · Reviewer_Wfb2 · 2023-11-23
> > > **Thanks for your responses**
> > >
> > > Thanks for your responses to my questions. Some of my concerns have been addressed. However, although the authors claimed DomainNet is a large-scale dataset, they only used five domains, which can not convince me about with the scalability of the proposed method.

---

> > > > ### Author Response · Authors · 2023-11-23
> > > > **Thanks for your reply**
> > > >
> > > > Thank you for your valuable comments and suggestions. While DomainNet has fewer continual learning stages, it boasts a larger number of classes and data samples in comparison to ImageNet-R or CIFAR-100. Additionally, DomainNet presents a notable domain gap between different classes, introducing difficulty for continual learning systems.
> > > >
> > > > We are open to incorporating additional datasets into our experiments to further address the concerns raised by the reviewer. If the reviewer could suggest a dataset for experimentation, we would be more than willing to include it in the final version of our work. Your insights and guidance are highly appreciated.

---

### Official Review · Reviewer_84n7 · 2023-11-08

**Soundness:** 3 good
**Presentation:** 2 fair
**Contribution:** 3 good
**Rating:** 6
**Confidence:** 3

**Summary:**

This paper proposes a single stage PCL framework that directly using the intermediate layer’s token embedding as a prompt query, so that the first query pre-training ViT could be avoid and half of the computational cost is avoid. The paper describe why using intermedia layer output to construct query embedding, and further propose QR loss to regulate the relationship between the prompt query and the prompt pool, and enhance the representation ability. Experimental result show that this approach could maintain performance under 50% less cost.

**Strengths:**

1. Reducing the computational cost by reducing query ViT seems well and could be regarded as new direction compared with prompt construction or weighting. I think the exploring on intermediate layer output as prompt is reasonable, organization of the paper is also good to follow.

2. The designed QR loss for supplementing the absent of [CLS] tokens in the query is interesting, it could maintain the representation power.

3. The result and discussion in the experimental part is convinced, the figure 4 shows the obvious improvement on consuming. Relative discussion and ablation study, parameter analysis also seems well.

**Weaknesses:**

I think some specific description should be more clear:
1. Although the training avoid the query ViT process, the author mentioned that QR loss need a reference ViT architecture. This process also need to forward the input, so for reducing the two-step forward, what's the difference between Query function forward and reference forward? Why the (training) computational cost still reduce 50% when QR loss with reference forward existing?
2. The author mention that this approach focuses on improving efficiency through a new query selection. But from the description, seems that how to select query that different from previous PCL is not clear, both of them apply the [CLS] token embedding (no requirement of query ViT in this process is the major difference).

**Questions:**

Is QR without query ViT could perform better generalization on unseen task/domain?

---

> ### Author Response · Authors · 2023-11-16
> **Rebuttal**
>
> We appreciate your positive feedback on our work. Please check our response to your questions and concerns.
>
> Q1: What is the difference between Query function forward and reference forward? Why the (training) computational cost still reduce 50% when QR loss with reference forward existing?
>
> A1: Sorry for the confusion. We would like to clarify QR loss is only used in OS-prompt++, which is an advanced version of our OS-Prompt. The OS-Prompt does not employ QR loss and is more efficient than existing two-stage methods during both training and inference.  The users can choose to use OS-prompt if they have fewer computing resources in training, but we highlight that OS-prompt can achieve comparable performance with existing SOTA. We summarize the difference between OS-Prompt and OS-Prompt++ in our general response.
>
> The Query function (used by previous PCL methods) is leveraged in training and inference, but the Reference function (proposed in this work) is only used in training, not inference. Therefore, with QR loss (i.e. OS-prompt++), the training computational cost does not reduce by 50%. We report the relative computational cost of both training and inference in Table 3. Please refer to the training cost for OS-prompt++ in Table 3.
>
> Q2: From the description, seems that how to select query that different from previous PCL is not clear, both of them apply the [CLS] token embedding.
>
> A2: We apologize for the unclear description.
> To obtain a query in layer l, prior methods utilize the [CLS] token from the 'last layer' of the Query ViT (Fig. 1, Left). In contrast, our approach involves using the [CLS] token from the  'layer l,' as depicted in Fig. 3 (arrow ②).
> Consequently, traditional PCL methods necessitate two feedforward steps: one for obtaining the query prompt and the second for the backbone ViT. In our OS-Prompt, we directly leverage the intermediate [CLS] token feature, which requires only one feedforward step.
> The difference in query selection strategy is also elaborated in Eq.(1) and Eq.(6) in our manuscript.
>
>
> Q3: Is QR without query ViT could perform better generalization on unseen task/domain?
>
> A3: We appreciate the insightful suggestion from the reviewer. This is an interesting and important concern because we use the intermediate layer’s [CLS] feature as a query, which might have less generalization power compared to the [CLS] feature from the last layer of query ViT.
>
> To this end, we conducted experiments on DomainNet [R1], a large-scale domain adaptation dataset consisting of around 0.6 million images distributed across 345 categories. Our continual domain adaptation task was created using five diverse domains from DomainNet: Clipart → Real → Infograph → Sketch → Painting. Thus every task has a different domain image, which is a proper experiment to check the generalization ability of our method. The experimental results are presented in the table below.
>
> | Method (Dataset: DomainNet) |    Acc (%)   | Forgetting score |
> |-----------------------------|:------------:|:----------------:|
> | UppderBound                 |     79.65    |         -        |
> | FT                          | 18.00 ± 0.26 |   43.55 ± 0.27   |
> | ER                          | 58.32 ± 0.47 |   26.25 ± 0.24   |
> | L2P                         | 69.58 ± 0.39 |    2.25 ± 0.08   |
> | Deep L2P                    | 70.54 ± 0.51 |    2.05 ± 0.07   |
> | DualPrompt                  | 70.73 ± 0.49 |    2.03 ± 0.22   |
> | CODA                        | 73.24 ± 0.59 |    3.46 ± 0.09   |
> | OS-Prompt                   | 72.24 ± 0.13 |    2.94 ± 0.02   |
> | OS-Prompt++                 | 73.32 ± 0.32 |    2.07 ± 0.06   |
>
> Our method (OS-prompt and OS-propmt++) achieves comparable performance compared to the previous PCL methods with ~50% inference computational cost. These findings suggest the adaptability of our method across various domains in different stages of continual learning.
>
> We have added the results in the Appendix.
>
> [R1] Peng et al. Moment matching for multi-source domain adaptation. ICCV19

---

### Official Review · Reviewer_PuDx · 2023-11-09

**Soundness:** 2 fair
**Presentation:** 3 good
**Contribution:** 2 fair
**Rating:** 5
**Confidence:** 4

**Summary:**

The submission proposes a method for one-stage prompt-based continual learning (PCL) where a separate forward pass of a ViT is not  required to extract the query tokens from the given image. Instead, they utilize intermediate [CLS] tokens from the previous layer as query tokens such that a single forward pass of the ViT is sufficient for PCL. To improve performance, they use a reference ViT during training to generate reference query tokens and use a query pool regularization loss to match the intermediate [CLS] tokens to the reference query tokens. Evaluation is done on CIFAR10 and ImageNet-R datasets, which show superiority of the proposed method.

**Strengths:**

- The problem of having to do 2 forward passes of a ViT in PCL is an important task to tackle.
- The proposed method is not specific to a model architecture, which makes it widely applicable.
- The idea itself is well presented and easy to follow.
- The empirical evaluation shows strong performance of the proposed method.

**Weaknesses:**

- The query pool regularization introduces extra computation in the training phase, especially as more advanced reference ViTs are used. I do not think this is particularly fair in comparison to prior work as they used less resources during training. Specifically, as the idea of continual learning depends much more heavily on the computation cost of training and inference due to the training phase being run continuously, I do not think matching the computation cost during the inference phase only is fully fair. Recent works such as [A] tackle this specific part of the resource constraints in continual learning and I believe the proposed method needs to discuss on the computation cost of the training phase more thoroughly.

[A] Real-Time Evaluation in Online Continual Learning: A New Hope, Y. Ghunaim et al., CVPR2023

**Questions:**

- Please see the weaknesses.

---

> ### Author Response · Authors · 2023-11-16
> **Rebuttal**
>
> We appreciate your constructive feedback on our work. Please check our response to your questions and concerns.
>
> Q1: The proposed method needs to discuss the computation cost of the training phase more thoroughly.
>
> A1:
> Thank you for the valuable feedback. We want to clarify that our OS-Prompt++ (with query pool regularization) does not result in increased training costs compared to previous Prompt Continual Learning (PCL) methods, such as CodaPrompt, DualPrompt, and L2P (refer to Table 3). Instead, OS-Prompt++ maintains the same training cost as the earlier approaches. On the other hand, our efficient version of PCL, named OS-prompt, minimizes the forward computation of the PCL framework, improving computational efficiency for both training and inference.
>
> In our work, we aim to solve 'Rehearsal-Free' continual learning, aiming to address both privacy and memory efficiency concerns, aligning with prior PCL research. Although our problem setting differs from online continual learning like [A], we agree with the reviewer that our proposed method needs a more thorough discussion of the training computation cost to provide a better understanding of the proposed approach. Following Table 1 provided in [A], we compute the relative training complexity with respect to ER or Experience Replay, which is the simple baseline with standard gradient-based training.
>
>
> | CL strategy       | Method             | Relative training complexity w.r.t ER |
> |-------------------|--------------------|---------------------------------------|
> | Experience Replay | ER                 | 1                                     |
> | Regularization    | LwF                | 4/3                                   |
> | Prompt-based      | L2P                | 1                                     |
> | Prompt-based      | DualPrompt         | 1                                     |
> | Prompt-based      | CodaPrompt         | 1                                     |
> | Prompt-based      | OS-Prompt (Ours)   | 2/3                                   |
> | Prompt-based      | OS-Prompt++ (Ours) | 1                                     |
>
> Here, we would like to clarify the training computational cost of prompt learning. In general neural network training,  the forward-backward computational cost ratio is approximately 1:2. This is due to gradient backpropagation ($\frac{dL}{da_l}=W_{l+1}\frac{dL}{da_{l+1}}$) and weight-updating ($\frac{dL}{dW_l}=\frac{dL}{da_l+1}a_l$), where $L$ represents the loss, $W$ denotes the weight parameter, $a$ is the activation, and $l$ is the layer index.  In prompt learning, the forward-backward computational cost ratio is approximately 1:1. This is in contrast to general neural network training, as only a small fraction of the weight parameters (less than 1\%) are updated.
>
>
> Bearing this in mind, we present the observations derived from the table, assuming that all methods employ the same architecture.
> * Previous PCL (L2P, DualPrompt, CodaPrompt) consists of two steps; (1) the query ViT requires only feedforward without backpropagation; (2) the backbone ViT feedforward-backward training with prompt tuning. This results in the relative training computational cost is 1 ($= \frac{Query ViT forward (1) + Backbone ViT forward (1) + Backbone ViT backward (1)}{ER forward (1) + ER backward (2)}$)
> * Similarly, our OS-Prompt++ also consists of two steps; (1) the reference ViT with only a feedforward step to calculate QR loss; (2) the backbone ViT feedforward-backward training with prompt tuning. This results in a relative training computational cost 1 ($= \frac{Reference ViT forward (1) + Backbone ViT forward (1) + Backbone ViT backward (1)}{ER forward (1) + ER backward (2)}$) with respect to ER.
> * On the other hand, our OS-Prompt requires one feedforward-backward step with prompt tuning. Therefore, relative training computaional cost becomes $\frac{2}{3}$ ($= \frac{Backbone ViT forward (1) + Backbone ViT backward (1)}{ER forward (1) + ER backward (2)}$). This shows the potential of our OS-Prompt on online continual learning.
>
> Overall, we propose two versions of the one-stage PCL method (i.e., OS-prompt and OS-prompt++), and these two options enable the users to select a suitable method depending on the problem setting. For example, for online continual learning, OS-prompt is a better option since it requires less training cost. On the other hand, for offline continual learning, one can use OS-prompt++ to maximize performance while spending more training energy.
>
> We have added this discussion in the Appendix.
>
> [A] Real-Time Evaluation in Online Continual Learning: A New Hope, Y. Ghunaim et al., CVPR2023

---

> > ### Comment · Reviewer_PuDx · 2023-11-23
> > **Author response**
> >
> > The reviewer appreciates the detailed response. However, the reference ViT used in OS-Prompt++ seems to consist of larger models, and authors even indicate that using larger models is beneficial. Thus, the presented argument about training costs do not fully convince me.

---

> > > ### Author Response · Authors · 2023-11-23
> > > **Reply to the author's comment**
> > >
> > > Thank you for your response. We would like to provide clarification regarding your comment. The reference ViT used in OS-Prompt++ does not consist of larger models; we employ the standard ViT-B/16, which serves as the backbone architecture. Additionally, the architectures utilized for ER, LwF, and other PCL methods are all based on ViT-B/16.
> > >
> > > As demonstrated in our response, despite the two-step feedforward during training (shared with other PCL methods), the total FLOPs are lower compared to other continual learning methods. Compared to prompt learning, the end-to-end training method (such as ER and LwF) incurs a threefold increase in computational cost due to weight updates for the entire set of parameters.
> > >
> > > In summary, our method employs the same model size as others, and prompt learning proves more training-efficient than other continual learning methods, primarily due to a lower number of parameter updates.

---

### Author Response · Authors · 2023-11-16
**General Response**

We’d like to thank all reviewers for their constructive feedback and suggestions on our work. We will address each reviewer’s questions and concerns point-to-point. And we welcome any further discussion on our paper. We have attached a modified PDF file based on the reviewer’s comment. A detailed description can be found in each rebuttal thread.


Here, we'd like to highlight the contribution of our method. In our paper, we propose two versions of the one-stage Prompt Continual Learning method (PCL); OS-prompt and OS-prompt++. OS-prompt minimizes the forward computation of the PCL framework, achieving comparable performance with less computational cost for both training and inference. OS-Prompt++ is the advanced version of OS-prompt with the additional reference ViT, improving accuracy by sacrificing training efficiency. We summarize the difference between OS-prompt and OS-prompt++  in the table below. There is an accuracy vs. training cost tradeoff between OS-prompt and OS-prompt++. These two options enable the users to select a suitable method depending on the computational budget and problem setting.

|    Method   |                    Architecture                   |        Loss        | Training Cost | Inference Cost | Accuracy |
|-----------|-------------------------------------------------|------------------|-------------|----------------|----------|
|  OS-Prompt  |                    Backbone ViT                   |       CE loss      |       ↓       | Same           |     ↓    |
| OS-Prompt++ | Backbone ViT + Reference ViT  (only for training) | CE loss  + QR loss |       ↑       | Same           |     ↑    |

---

> ### Author Response · Authors · 2023-11-22
> **General Response  (continue)**
>
> We sincerely appreciate the constructive feedback and valuable suggestions provided by the reviewers for our work. We are committed to addressing each of the reviewers' questions and concerns in a point-to-point manner.
>
> To facilitate this process, we have revised our manuscript and have attached the modified PDF file based on the reviewers' comments. Since we have a day for Author/Reviewer discussion, we welcome any further discussion and would be grateful.

---

### Comment · Area_Chair_B7Sy · 2023-11-22
**less than one day**

Dear Reviewers,

If you have already responded to authors rebuttal, Thank you! If not, please take some time, read their responses and acknowledge by replying to the comment. Please also update your score, if applicable.

Thanks everyone for a fruitful, constructive, and respectful review process.

Cheers, Your AC!

---

### Meta-Review · Area_Chair_B7Sy · 2023-12-05

**Metareview:**

This paper introduces a method improve the efficiency of the conventional two-stage PCL by repurposing the intermediate
token embeddings for prompts. To compensate for the small drop in performance it introduces a regularization loss and runs evaluation on class-incremental continual learning scenarios on cifar-100 and imagenet-r datasets.

Strengths:
- Well motivated problem of reducing the computation during inference for continual learning
- Simple yet effective use of previous layer token embeddings

Weaknesses:
-  Lack of proper justification and analysis: The token embedding reuse itself sounds like a regularization scheme where token embeddings are penalized to stay close to the previous ones. Its relation to QR loss is also worth deeper dive. Overall, why and how the method works is not discussed/analyzed properly.
- Applicability and generality of the findings: Issues include increased computation cost due to query pool regularization, scalability uncertainties with larger datasets, and a lack of detailed exploration into the limitations of the introduced QR loss. Improving the method's broader relevance and addressing these concerns needs another iteration of the manuscript.

**Justification For Why Not Higher Score:**

The following shortcomings of the paper remained unresolved and need an in-depth revision:
1) lack of proper justification and understanding,
2) applicability and scalability of the results,
3) limitation and relation to the baselines.

**Justification For Why Not Lower Score:**

N/A

---

### Decision · Program_Chairs · 2024-01-16

Reject